# Protective effect conferred by prior infection and vaccination on COVID-19 in a healthcare worker cohort in South India

**Malathi Murugesan**[1], **Prasad Mathews**[2], **Hema Paul**[1], **Rajiv Karthik**[3], **Joy John Mammen**[4], **Priscilla Rupali**[3]*

1 Department of Clinical Microbiology & Hospital Infection Control Committee, Christian Medical College, Vellore, Tamil Nadu, India, 2 Department of Geriatrics & Hospital Infection Control Committee, Christian Medical College, Vellore, Tamil Nadu, India, 3 Department of Infectious Disease & Hospital Infection Control Committee, Christian Medical College, Vellore, Tamil Nadu, India, 4 Department of Transfusion Medicine, Christian Medical College, Vellore, Tamil Nadu, India

* prisci@cmcvellore.ac.in

## Abstract

### Background

The emergence of newer variants with the immune escape potential raises concerns about breakthroughs and re-infections resulting in future waves of infection. We examined the protective effect of prior COVID-19 disease and vaccination on infection rates among a cohort of healthcare workers (HCW) in South India during the second wave driven mainly by the delta variant.

### Methods and findings

Symptomatic HCWs were routinely tested by RT-PCR as per institutional policy. Vaccination was offered to all HCWs in late January, and the details were documented. We set up a non-concurrent cohort to document infection rates and estimated protective efficacy of prior infection and vaccination between 16th Apr to 31st May 2021, using a Cox proportional hazards model with time-varying covariates adjusting for daily incidence. Between June 2020 and May 2021, 2735 (23.9%) of 11,405 HCWs were infected, with 1412, including 32 re-infections, reported during the second wave. 6863 HCWs received two doses of vaccine and 1905 one dose. The protective efficacy of prior infection against symptomatic infection was 86.0% (95% CI 76.7%–91.6%). Vaccination combined with prior infection provided 91.1% (95% CI 84.1%–94.9%) efficacy. In the absence of prior infection, vaccine efficacy against symptomatic infection during the second wave was 31.8% (95% CI 23.5%– 39.1%).

### Conclusions

Prior infection provided substantial protection against symptomatic re-infection and severe disease during a delta variant driven second wave in a cohort of health care workers.

**Data Availability Statement:** All data are fully available without restriction. The raw data base and codes used for the study is uploaded in the data

repository Dryad (https://doi.org/10.5061/dryad.n8pk0p2x8).

**Funding:** The author(s) received no specific funding for this work.

**Competing interests:** The authors have declared that no competing interests exist.

## Introduction

Severe acute respiratory syndrome coronavirus-2 (SARS-CoV-2) infected 27 Million people and caused 0.3 Million deaths till May 2021 in India [1]. The dramatic increase in the number of cases during the second wave of the COVID-19 pandemic was attributed to multiple factors like a large susceptible unvaccinated population, lack of durability of an immune response, variants and mutants with their enhanced disease transmission and virulence capabilities [2]. SARS-CoV2 infection produces variable levels of antibodies in humans based on the nature of infection i.e. asymptomatic or symptomatic, the severity of illness, and the host's immune response [3, 4]. Most studies published on humoral antibody responses to SARS-CoV2 infection have shown that the persistence of spike protein antibodies in COVID-19 infected persons range from 3 to 10 months [5, 6]. In a study conducted among health care workers (HCW) in Oxford university hospitals, United Kingdom, there were no symptomatic infections reported among HCW who had detectable levels of anti-spike antibodies [7]. As the duration of antibody persistence differs across the population, the risk of re-infection remains a concern. In addition, it is unclear whether the presence of anti-spike antibodies can be correlated with the presence of protective neutralising antibodies. COVID-19 re-infection in a previously infected individual can occur either due to decay in the antibody response or due to emerging new variants or mutants. The emerging mutants have raised concerns globally with respect to their transmission rates, the severity of infection and escape from immunity [8]. SARS-CoV2 variants of concern identified so far, have shown either slightly reduced or a potential reduction in neutralisation by post-vaccination sera [8–10]. The newer variants circulating in India, namely B.1.617, B.1.617.1, B.1.617.2, B.1.617.3, and now B.1.529, have been of great concern as they can evade neutralising antibodies [11], thereby causing re-infections among previously infected persons causing breakthrough infections in vaccinated individuals. Evidence on the durability of the protective immune response after COVID-19 vaccination and prior infection is still evolving. In addition, studying the protective effect of vaccination in a population already exposed or infected will not reflect the actual contribution of protective effect that a previous infection confers or adds to vaccination. Hence, we conducted this study to look at the rate of COVID-19 re-infections and the protective effect of a previous COVID-19 infection among a cohort of HCW in a tertiary care institution located in South India.

## Materials and methods

This study was approved by the institutional ethics committee and research board (IRB Min no. 13980). This non-concurrent cohort study was conducted among the staff of Christian Medical College, Vellore, a 2600 bedded tertiary care institution located in South India, and examined the incidence and predictors of infection during the second wave. Students and temporary workers were excluded from the study as they were not on the payroll database and were not required to be at the institution during the study period. Participant data has been anonymized and administrative approval for sharing the deidentified HCW data has been obtained. Hence our institution's ethical committee has waived consent requirements. As per institutional protocol, employees were required to monitor symptoms and report to the staff health services if they developed symptoms of COVID-19, where a nasopharyngeal specimen was obtained and tested for SARS-COV2 using an RT-PCR assay (RealStar® SARS-CoV2 RT-PCR kit 1.0, Altona Diagnostics). Since staff are offered comprehensive healthcare free of cost, all HCW with COVID-19 infections were treated in our institution. All staff who undergo COVID testing are registered through a single portal and positive results are captured from the laboratory registry. Health care worker found to have Influenza Like Illness (ILI) symptoms i.e., fever and cough ± ILI symptoms) or Severe Acute Respiratory Infection (SARI)

symptoms i.e., (fever, cough and breathlessness) with laboratory confirmed COVID-19 infection were defined as having previous infection. HCW who developed a laboratory confirmed COVID-19 infection 12 weeks from the date of first positive infection were considered to have a re-infection. The demographic, clinical and exposure variables and vaccination history were prospectively documented in an electronic database from all those presenting for COVID testing. The severity of COVID-19 infection was assessed by the WHO severity scale (May 2020) [12]. All patient related data were abstracted from medical records.

In late January 2021, the institution organised a systematic effort to vaccinate all staff against SARS-COV2. The vaccines given were ChAdOx1 nCoV-19 Corona Virus Vaccine COVISHIELD™ and the whole virion inactivated BBV152 vaccine COVAXIN™. All immunisation was documented along with the date of vaccination, type of vaccine and any adverse events. Linking the SARS-COV2 testing data set with the vaccination and administrative payroll information, we established a non-concurrent cohort that included all current employees. Every employee has an unique employment ID which was used to match across the datasets. Two investigators independently assessed the datasets to verify the accuracy of the data and linkages between the datasets. Cases of SARS-COV2 that were detected before 1st April 2021 were considered as the cases that occurred prior to the second wave. Cases emerging between 1st April 2021, and 31st May 2021, were part of the large second wave that coincided with the emergence of the Delta variant. Participants were considered fully immunised 14 days after the second dose of the vaccine and partially immune 21 days after the first dose of the vaccine. Since most employees received their vaccination just before the second wave's onset and were considered immune by 16th April 2021, we used this date for entry into a survival analysis and to establish the baseline risk status.

## Statistical analysis

Those who were either unvaccinated or had not completed 14 days after the second dose by entry, were considered unvaccinated. Anyone who reported a RT-PCR confirmed SARS-COV2 infection previously were considered to have been previously infected. Participants were categorised into four risk groups based on their prior infection and vaccination status, namely, the unvaccinated and previously uninfected; vaccinated and previously uninfected; unvaccinated and previously infected; and vaccinated and previously infected. A sensitivity analysis that excluded participants who had received a single dose was not significantly different from the one that included those received one dose as unvaccinated. Hence the binary classification of vaccinated and unvaccinated was based on the completion of two doses of vaccination 2 weeks after the second dose.

Kaplan Meier Survival analysis was done with failure defined as the acquisition of infection during the analysis period. A Log-rank test was performed to compare the survival curves across the four risk groups. We developed a Cox-proportional hazards (PH) model with time-varying covariates adjusting for smoothed daily incidence of COVID-19 and potential confounders (S1 Table). The model included participant age, type of work, sex, history of prior infection and vaccination, as epidemiologically relevant factors. The model was tested for the proportional-hazards assumption on Schoenfeld's residuals and the PH assumption was not violated (p value—0.134). Efficacy of prior infection and vaccines to prevent symptomatic infection in the study period were calculated as VE = 1- hazard ratio from the Cox proportional hazard model. All data analysis was performed using Stata 15.1 (Statacorp LLC, College Station, TX).

**Table 1. Demographic details of the health care workers cohorted based on COVID results.**

| Characteristics | Positive cohort (n = 2735) | Negative cohort (n = 8670) | All health care workers (n = 11405) |
|---|---|---|---|
| **Gender** | | | |
| Male | 1094 (40.0%) | 3580 (41.3%) | 4674 (41.0%) |
| Female | 1641(60.0%) | 5090 (58.7%) | 6731 (59.0%) |
| **Age** | | | |
| Less than 30 years | 3078 (35.5%) | 868 (31.7%) | 3946 (34.6%) |
| 30 to 39 years | 2955 (34.1%) | 967 (35.4%) | 3922 (34.4%) |
| 40 to 49 years | 1785 (20.6%) | 621 (22.7%) | 2406 (21.1%) |
| 50 to 59 years | 807 (9.3%) | 277 (10.1%) | 1084 (9.5%) |
| 60 years or older | 45 (0.5%) | 2 (0.1%) | 47 (0.4%) |
| Median (IQR) | 33.7 (27.7–42.4) | 34.6 (28.5–42.9) | 33.9 (27.8–42.5) |
| **Professional category** | | | |
| Consultant doctor | 146 (5.34%) | 664 (7.66%) | 810 (7.10%) |
| Trainee doctor | 243 (8.88%) | 947 (10.92%) | 1190 (10.43%) |
| Nursing | 1031 (37.70%) | 2806 (32.36%) | 3837 (33.64%) |
| Technician | 232 (8.48%) | 833 (9.61%) | 1065 (9.34%) |
| Pharmacist | 87 (3.18%) | 262 (3.02% | 349 (3.06%) |
| Attendant | 424 (15.50%) | 1271 (14.66%) | 1695 (14.86%) |
| Clerical staff | 93 (3.40%) | 246 (2.84%) | 339 (2.97%) |
| Support staff/others | 479 (17.51%) | 1641 (18.93%) | 2120 (18.59%) |

## Results

Between 1st June 2020 to 31st May 2021, 11405 health care workers were on the payroll. 41% of the study population are males. The median [IQR] for age in the study cohort is 33.9 years [27.8–42.5 years] (S1 Fig). Among the professional categories, 33.64% were nurses, 18.59% support staff, 17.53% doctors, 14.86% hospital attendants, 9.34% technicians, 3.06% pharmacists and 2.97% clerical staff (Table 1). By 31st May 2021, 2735 (23.9%) developed COVID-19 infection (Fig 1). 1355 HCW were infected with COVID-19 infection prior to second wave and 1380 in the second wave. Assessing the severity of COVID-19 infections, 98.0%, 0.7% and 1.3% belonged to mild, moderate and severe/critical categories in the first wave and 99.44%, 0.2% and 0.4% in the second wave, respectively.

During the second wave, 32 out of 1355 (2.36%) previously infected HCW were re-infected, including 28 between 16th April and 31st May 2021. The median duration from the first infection to the second infection was 258.5 days. All 32 cases of re-infection were mild. One patient with re-infection was hospitalised for an unrelated indication. Fourteen of the 32 re-infected cases had received two doses of vaccination at least two weeks before being detected to have COVID-19.

The vaccination campaign from January 2021 successfully vaccinated 8768 HCW (76.9%) by 31st May 2021. The vaccines used were ChAdOx1 nCoV-19 Corona Virus Vaccine COV-ISHIELD™ (94.23%) and the whole virion inactivated BBV152 vaccine COVAXIN™ (5.77%). Among the vaccinated individuals, 1905 received one dose while 6863 had completed two doses of vaccination. The temporal trends of COVID-19 infection and vaccination are presented in Fig 2. Between 16th April 2021 to 31st May 2021, the cumulative incidence risk of COVID-19 infection among those previously uninfected was 14.9% if unvaccinated and 11.1% when vaccinated, respectively. Among those previously infected, the cumulative incidence in

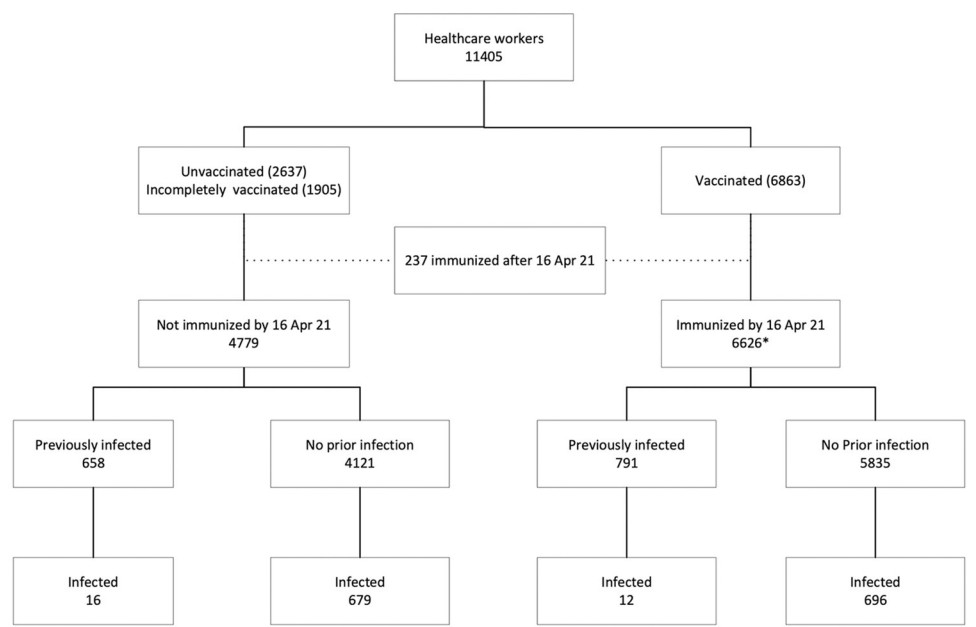

**Fig 1. Flow diagram.** * received two doses of vaccine by 2<sup>nd</sup> April 2021.

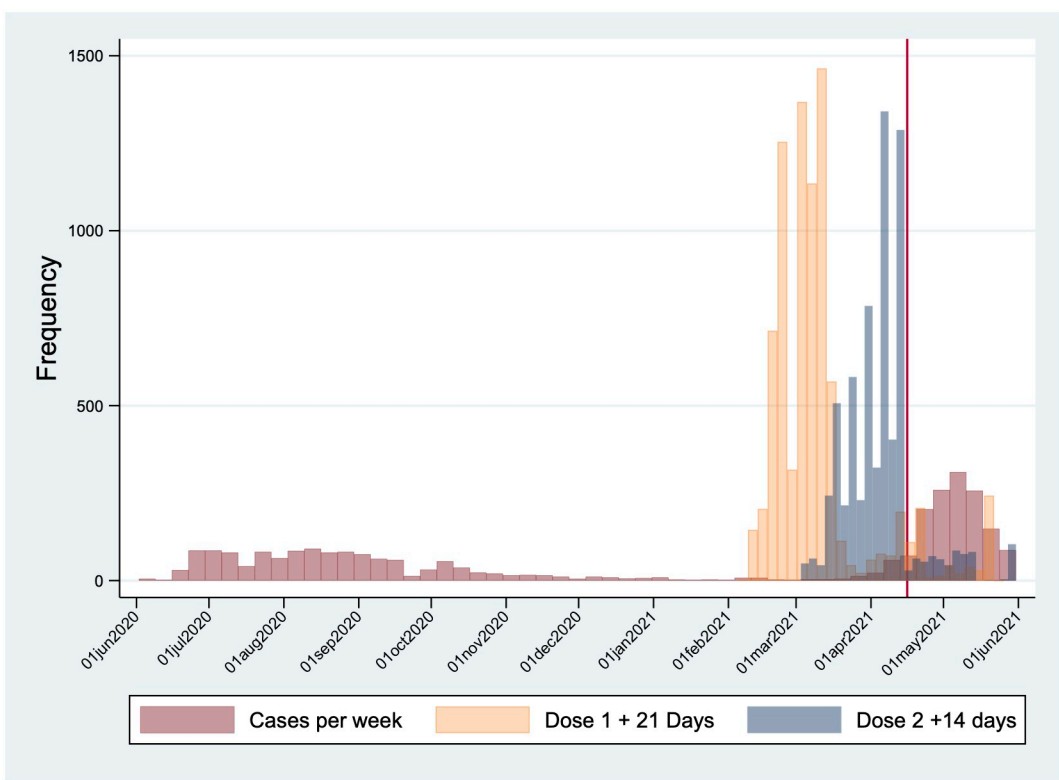

**Fig 2. Temporal trends of COVID-19 infection and vaccination in our study cohort.**

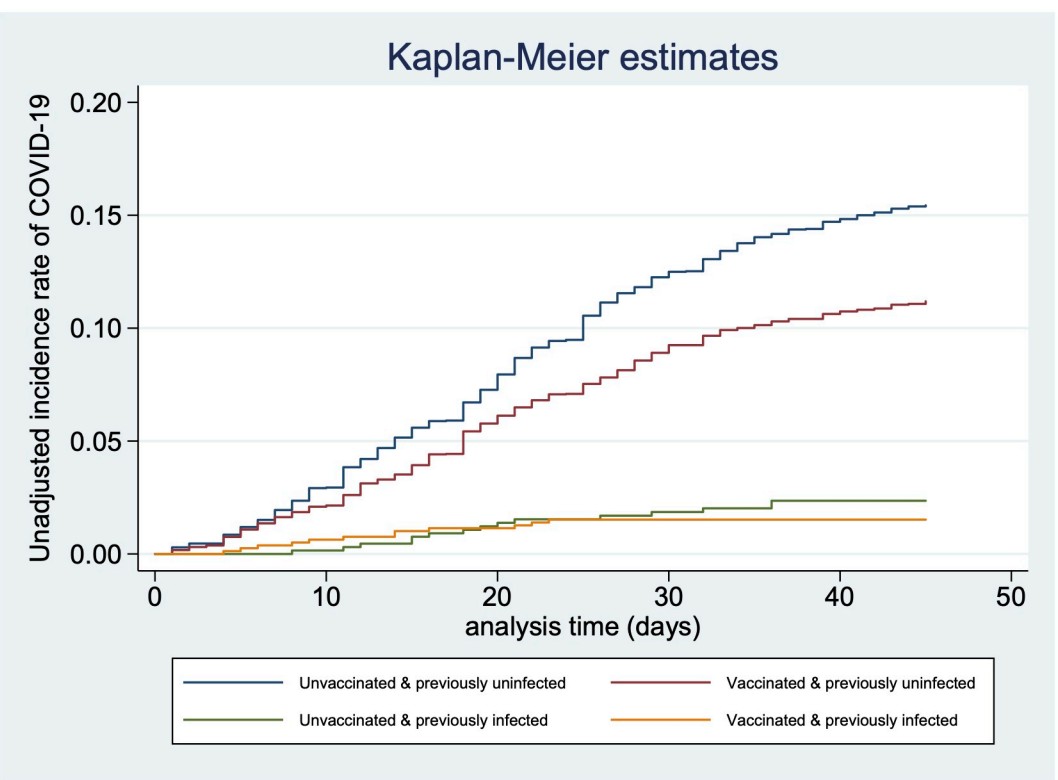

**Fig 3. Incidence rates of COVID-19 based on vaccination and prior infection.**

the unvaccinated was 2.1% and 1.4% in the vaccinated (S2 Fig). The Kaplan Meier survival curve in Fig 3 compares the unadjusted incidence rates across vaccination and prior infection status. All those included in the analysis were present from the beginning of the study to the end. There was no mortality during the study period.

The protective efficacy of prior infection against symptomatic infection was 86.0% (95%CI 76.7%–91.6%). Vaccination combined with prior infection provided 91.1% (95%CI 84.1%–94.9%) efficacy. In the absence of prior infection, vaccine efficacy against symptomatic infection during the second wave was 31.8% (95%CI 23.5%– 39.1%). The minimum interval between two doses of vaccine was one month but varied among different individuals. Sample size was inadequate to look at VE for varied intervals.

Of the 2760 episodes of COVID-19, 35 (1.3%) were moderate to severe episodes. The episodes during the second wave were milder with only 8 of 1412 episodes (0.57%) being classified as moderate or severe as compared to 27 of 1355 episodes (2.0%) prior to second wave.

## Discussion

In this large single-centre cohort study of HCW, 32 were re-infected (2.36%) in the second wave. A systematic review conducted by SeyedAlinaghi *et al.*, 2020 evaluated 31 studies, of which only 8 reported re-infections in a previously infected and recovered group of patients [13]. Another large scale multicentric study on health care workers (SIREN study) published recently by Hall *et al.*, 2021 in the United Kingdom that showed the incidence risk ratio for all re-infections was 0.159 compared with PCR-confirmed primary infections [14]. The re-infection rates in our study population is slightly higher than reported by SIREN study, probably

due to the B.1.617.2 delta variant predominance in India during the second wave. An invitro neutralisation study performed with sera from the convalescent and vaccinated individuals, revealed that the delta variant B.1.617.2 has increased resistance to neutralisation in sera from convalescent individuals who were unvaccinated vs those who were vaccinated [11, 15].

In our study, the estimates for overall protection after COVID-19 infection against a repeat infection was 86.0% which is supported by other studies conducted in UK, Denmark, and the USA [14, 16, 17]. The protective efficacy of COVID-19 vaccination in various studies is reported to be between 60 to 90% based on the type of vaccine and the doses received. However, this efficacy is likely to be the result of a robust immune response contributed to both by the previous infection and vaccination and not the vaccine alone. Our study looked at a captive population of HCW of age group 18 to 85 years who were closely monitored for development of COVID-19 infections both before and after vaccination. With an adequate testing facility and appropriate documentation of COVID-19 infection, we were able to explore the protective effect of a previous COVID-19 infection. Hence the high efficacy of vaccination of 91.1% is likely to be due to the combined protective effect of previous infection and vaccination together rather than vaccination alone.

We found a lower severity of COVID-19 infection during the second wave 0.4% as compared to 1.3% in the first wave. Age did not seem to influence the risk of acquiring infection in our study as the median age of health care workers in our cohort was 33.9 years. This younger age was in variance with other large health care cohort study (SIREN) reported from developed countries which reported a median of 45.7 years [IQR 35·4–53·5] but in keeping with that reported in general population and in HCW in India [18, 19].

A recently published study from England has shown that the adjusted vaccine efficacy of ChAdOx1 nCoV-19 two doses against alpha variant was 74.5% (95% CI 68.4 to 79.4) but drops against delta variant with one dose and two doses offering 30.0% (24.3–35.3) and 67.0% protection (61.3–71.8) respectively [20]. Even though, the protective efficacy of vaccination against the delta variant differs based on the type of vaccine and study population, the impact of vaccination in bringing down the hospitalisation rates and severity of illness is of utmost importance.

This study supports the premise that a robust immune response is produced by a natural COVID-19 infection and an additional protective effect is contributed to by vaccination. This maybe important to inform public health policies for optimisation of vaccination campaigns targeting the HCW and then unaffected epidemiological areas first followed by the recently affected hotspots during the second wave. As the immunity lasts for 3–10 months [5, 6], mass vaccination should be targeted at districts/states with a lower rate of seroconversion on a priority basis before newer variants emerge.

## Limitations

We noted several limitations. Firstly, we were unable to use the standard definition of re-infections, as we do not do routine genomic sequencing for all infections, vaccine breakthroughs or even re infections. Hence all these cases were considered as probable re-infections if infections occurred greater than 12 weeks from a previous infection with a Ct value <33 in all those considered re infected. Secondly, it is unclear whether these figures could be extrapolated to the community as HCW as a population are at a constantly higher risk of exposure and in our hospital belonging to a younger age group. Since our hospital policy mandated compulsory testing, if symptomatic, it is likely that we detected a higher rate of infection [21, 22]. Thirdly, though HCW with symptoms or a close contact with a COVID-19 case were tested, we probably missed asymptomatic infections. Hence assessment of vaccine efficacy may have been

confounded by prior asymptomatic infection that was undetected. This could have impacted the vaccine efficacy in either direction. It could have decreased the incidence of a second infection, in those labelled as uninfected or it could have potentiated the protection of vaccinated individuals who may have had undetected asymptomatic previous infection.

## Conclusion

Natural COVID-19 infection likely produces a robust immune response with 86% protection against re infection. However, when previously infected individuals are vaccinated, the protective efficacy goes up to 91%. Vaccination should be encouraged irrespective of a prior COVID-19 infection.

## Supporting information

**S1 Fig. Age histogram of the study cohort.**
(TIF)

**S2 Fig. Unadjusted incidence of symptomatic COVID-19 infection by vaccination and previous infection status.**
(TIF)

**S1 Table. Cox-proportional hazards (PH) model.**
(DOCX)

## Acknowledgments

The authors sincerely thank Dr Jacob John, Professor of Community Medicine, Christian Medical College, Vellore for his various methodological inputs and statistical expertise.

## Author Contributions

**Conceptualization:** Malathi Murugesan, Prasad Mathews, Priscilla Rupali.

**Data curation:** Malathi Murugesan, Hema Paul, Rajiv Karthik, Joy John Mammen, Priscilla Rupali.

**Formal analysis:** Malathi Murugesan, Hema Paul, Rajiv Karthik, Joy John Mammen, Priscilla Rupali.

**Methodology:** Malathi Murugesan, Prasad Mathews, Hema Paul, Priscilla Rupali.

**Project administration:** Prasad Mathews, Joy John Mammen, Priscilla Rupali.

**Resources:** Malathi Murugesan, Prasad Mathews.

**Supervision:** Prasad Mathews, Rajiv Karthik, Joy John Mammen, Priscilla Rupali.

**Validation:** Malathi Murugesan, Priscilla Rupali.

**Writing – original draft:** Malathi Murugesan, Hema Paul.

**Writing – review & editing:** Prasad Mathews, Rajiv Karthik, Joy John Mammen, Priscilla Rupali.

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
