## [Decision Letter · Decision Letter 0]

31 Jan 2022

PONE-D-21-35055Protective effect conferred by prior infection and vaccination on COVID-19 in a Healthcare Worker Cohort in South IndiaPLOS ONE

Dear Dr. Rupali,

Thank you for submitting your manuscript to PLOS ONE. After careful consideration, we feel that it has merit but does not fully meet PLOS ONE’s publication criteria as it currently stands. Therefore, we invite you to submit a revised version of the manuscript that addresses the points raised during the review process.

Your manuscript was reviewed by 2 experts in the field. Both identified important problems in your submission. Please review the attached comments and provide point-by-point responses.

We look forward to receiving your revised manuscript.

Kind regards,

Yury E Khudyakov, PhD

Academic Editor

PLOS ONE

Journal Requirements:

4. Please note that supplementary tables (should remain/ be uploaded) as separate "supporting information" files.

Reviewers' comments:

Reviewer's Responses to Questions

**Comments to the Author**

1. Is the manuscript technically sound, and do the data support the conclusions?

Reviewer #1: Partly

Reviewer #2: Yes

2. Has the statistical analysis been performed appropriately and rigorously? 

Reviewer #1: I Don't Know

Reviewer #2: Yes

3. Have the authors made all data underlying the findings in their manuscript fully available?

Reviewer #1: No

Reviewer #2: No

4. Is the manuscript presented in an intelligible fashion and written in standard English?

Reviewer #1: Yes

Reviewer #2: Yes

5. Review Comments to the Author

Reviewer #1: The paper addresses an important question and the data is relevant, but several questions still need to be addressed before considering publication.

1/ More detailed informations is needed on the patient caracteristics with respect to target population, e.g. is the cohort younger/older, has it more co-morbidities, etc; unless I missed a table, this data is simply absent. Please give mean/median, min/max/quartiles,std (where applicable) etc. information. This is even more a requirement given the Cox analysis latter that

includes e.g. age as confounder.

2/ More data would be welcome to quantify the severity of the infection: do you have any viral load data in order to quantify it ? Or maybe the number of PCR cycles ...

3/ In principle everybody is convinced that initial infection provides a protection against a challenge (secondary infection); in order to be useful the work needs to give more quantitative insights, for instance is there any data available to correlate, in a quantitative manner protection with antibody level following a previous infection ?

Any other similar data welcome.

3/ Technical question: I'm not sure I understood the Cox results in suppl. table 1: for instance, for age HR turns out to be 1.0, or I would expect the age to have a really visible impact. Please comment this into the discussion section.

4/ On the other hand, if (see below) the age in the cohort is really so centered in the 30-40years region, maybe age is not so important in this range. Please provide a FULL age distribution histogram.

5/ This should be listed more clearly as a limitation of the study. Just to be sure, can you positively confirm that age and the other confounders in suppl. table 1. were indeed considered as parts of a MULTIVARIATE Cox analysis ?

6/ Also, I see no p-values in the results ... Or is maybe "standard error" column in fact a "p-value" column ?

Small typo (in many places) and remarks:

- replace "cox" by "Cox".

- in the table 1 "Vaccinated Infected 0.90" I guess the value is not "0.90" but rather "0.10" or "0.09" ; please check very thoroughly

- second line in "Results": what does "median IQR for age" means ? is 33.9 the median and 27.8 Q1 and 42.5 the Q3 ? If so, say so, the actual formulation is incorrect, a minimalistic change is to write "median (IQR)"

- is it possible to provide data in a more detailed manner (e.g. as a supplementary online data file) ?

Reviewer #2: The authors present a well studied HCW cohort and looked at the protective effect of vaccination and prior infection. I think it is a very interesting study, and a unique cohort to untangle the effects of vaccination vs prior infection. It was very well written and very clearly explained.

The limitations of the study are all acknowledged, but could have been expanded further in the discussion as to how they effect the results. For example, you mention in the limitations you could only look at symptomatic infections, but what impact might this have on your results? i.e., your reference group is unvaccinated, no prior infection - but some of those may have in fact had asymptomatic infections, so how might that influence your results? The same with those who are vaccinated, no prior infection. You could even consider doing a sensitivity to see what effect it would have if XX percent of those who report as not infected were in fact previously infected. This is only a suggestion, and I do not think it is an absolute requirement, but I do think more discussion on this limitation would help readers better understand the context of these results. You also mention about the fact that HCW are likely to have repeated exposures - but how might that effect the results?

Furthermore, you report a much lower vaccine efficacy in this study compared to previous studies, and I think this could be expanded on a little in the discussion. Why do you believe that vaccine efficacy (no prior infection) was so much lower- was this because you looked at it in isolation? And is this different to how other studies have looked at vaccine efficacy? And what effect does the fact you've looked HCW (with repeated exposures) have (if any_?

Minor points:

I thought the methods were very well written and clear. I just found the description of the cohort study a little confusing at times understanding the start and end dates. From my understanding you had a cohort between 16th April to 30th/31st May (second wave) 2021 and included all HCW. You then looked at infection events during this wave. Might be useful to have a sentence summing this up at the end of your descriptions of the cohort. (Also, minor point you mention 30th May in abstract but 31st in methods).

I think in Supplementary table 1 it should maybe say "prior infection" rather than infection, and it could be a little confusing. I also think the reference group (which I believe is the unvaccinated, no prior infection) should be clearly stated - perhaps below the table.

I think there's a mistake in Supplementary table 1 - the 95% confidence intervals for "vaccinated infected" group are smaller than the hazard ratio?

There are some details missing from the methods (which are mentioned in the discussion) but I think it would help to have them in the methods:

- Type of testing done, and a clearer explanation of who was eligible (symptomatic and if you have a positive household contact?

- Vaccine type

Previous infection (what was classified as a previous infection)

6. PLOS authors have the option to publish the peer review history of their article (what does this mean?). If published, this will include your full peer review and any attached files.

Reviewer #1: No

Reviewer #2: No

---

## [Author Response · Author response to Decision Letter 0]

18 Feb 2022

RESPONSE TO REVIEWER COMMENTS & QUERIES

Manuscript ID

PONE-D-21-35055

Reviewer #1: 

Major points:

The paper addresses an important question and the data is relevant, but several questions still need to be addressed before considering publication. 

1. More detailed information’s is needed on the patient characteristics with respect to target population, e.g. is the cohort younger/older, has it more co-morbidities, etc; unless I missed a table, this data is simply absent. Please give mean/median, min/max/quartiles, std (where applicable) etc. information. This is even more a requirement given the Cox analysis latter that includes e.g. age as confounder. 

Authors` response: We thank you for your suggestions. As mentioned by the reviewer, age of the target population was already present in the manuscript, however for clarity, we have added the patient characteristics (Age stratification, professional category and gender) in table 1.

2. More data would be welcome to quantify the severity of the infection: do you have any viral load data in order to quantify it ? Or maybe the number of PCR cycles.

Authors` response: The Indian Council of Medical Research, did not recommend cycle threshold (ct) values be used as an indicator of clinical severity as many sampling or interpretation (pre-analytical and analytical) factors can influence the ct values. Therefore, viral loads are not a suitable correlate of severity of infection. Hence, we have provided data based on the WHO clinical severity scale which is a clinically robust assessment of baseline clinical status and likely to be far more accurate. 

Reference:https://www.icmr.gov.in/pdf/covid/techdoc/Advisory_on_correlation_of_COVID_severity_with_Ct_values.pdf

3. In principle everybody is convinced that initial infection provides a protection against a challenge (secondary infection); in order to be useful the work needs to give more quantitative insights, for instance is there any data available to correlate, in a quantitative manner protection with antibody level following a previous infection ?

Any other similar data welcome. 

Authors` response: At the point when this data was collected, there was inadequate recognition of the impact of prior infection on subsequent infection (study was submitted in November 2021). This study was uniquely able to establish based on RT-PCR confirmed infections, that previously infected individuals were protected from delta infection independent of the vaccination status. We believe that this is mediated in a large part by the cellular immunity in addition to the humoral response to an exposure. Quantitative antibody assays were not performed in this cohort and are not recommended as per CDC. 

Reference: https://www.cdc.gov/coronavirus/2019-ncov/lab/resources/antibody-tests-guidelines.html

4. Technical question: I'm not sure I understood the Cox results in suppl. table 1: for instance, for age HR turns out to be 1.0, or I would expect the age to have a really visible impact. Please comment this into the discussion section. 

Authors` response: Age does not seem to influence the risk of acquiring infection in our study as the median age of health care workers in our cohort was 33.9 years which is a relatively younger age group. However, we agree with the reviewer that it has been shown in peer reviewed literature that older age does correlate with an increased risk of acquisition of infection and progression to severe disease. 

We have included this in the discussion section.

5. On the other hand, if (see below) the age in the cohort is really so centered in the 30-40years region, maybe age is not so important in this range. Please provide a FULL age distribution histogram. 

Authors` response: Age distribution has been provided in the table 1. A histogram has been provided as a supplementary material (figure 2).

6. This should be listed more clearly as a limitation of the study. Just to be sure, can you positively confirm that age and the other confounders in suppl. table 1. were indeed considered as parts of a MULTIVARIATE Cox analysis ? 

Authors` response: We do not think the fact that age was not correlating with the risk of acquisition of infection is a limitation, we feel that this is so because of the predominantly younger cohort in our study. The raw data has been shared already and we are happy to share the codes for statistical analysis.

7. Also, I see no p-values in the results ... Or is maybe "standard error" column in fact a "p-value" column ? 

Authors` response: The SE column is not a p value column. We did not add the p value as it is a non-randomized comparison and it would be statistically inappropriate to provide p values here. However, we can remove the SE column and replace with the p value column if reviewer feels that is necessary. 

Minor points:

8. Small typo (in many places) and remarks - replace "cox" by "Cox".

Authors` response: We have corrected the typo.

9. in the table 1 "Vaccinated Infected 0.90" I guess the value is not "0.90" but rather "0.10" or "0.09" ; please check very thoroughly

Authors` response: We thank you for pointing out the error. The hazard ratio is 0.09 (95 % CI 0.05-0.159). We have corrected it in the manuscript.

10. second line in "Results": what does "median IQR for age" means ? is 33.9 the median and 27.8 Q1 and 42.5 the Q3 ? If so, say so, the actual formulation is incorrect, a minimalistic change is to write "median (IQR)"

Authors` response: The changes has been made as median (IQR).

11. is it possible to provide data in a more detailed manner (e.g. as a supplementary online data file) ? 

Authors` response: Yes we can provide the dataset. We have uploaded the data and codes in data repository (Dryad - https://doi.org/10.5061/dryad.n8pk0p2x8) and its under private view.

Reviewer #2: 

Major points:

1. The authors present a well-studied HCW cohort and looked at the protective effect of vaccination and prior infection. I think it is a very interesting study, and a unique cohort to untangle the effects of vaccination vs prior infection. It was very well written and very clearly explained. The limitations of the study are all acknowledged, but could have been expanded further in the discussion as to how they effect the results. For example, you mention in the limitations you could only look at symptomatic infections, but what impact might this have on your results? i.e., your reference group is unvaccinated, no prior infection - but some of those may have in fact had asymptomatic infections, so how might that influence your results? The same with those who are vaccinated, no prior infection. You could even consider doing a sensitivity to see what effect it would have if XX percent of those who report as not infected were in fact previously infected. This is only a suggestion, and I do not think it is an absolute requirement, but I do think more discussion on this limitation would help readers better understand the context of these results. You also mention about the fact that HCW are likely to have repeated exposures - but how might that effect the results? 

Authors` response: We thank the reviewer for a positive feedback and suggestions. As mentioned by the reviewer, the assessment of vaccine efficacy may have been confounded by prior asymptomatic infection that was undetected. This could have impacted the vaccine efficacy in either direction. It could have decreased the incidence of a second infection, in those labelled as uninfected or it could have potentiated the protection of vaccinated individuals who may have had undetected asymptomatic previous infection. It is therefore difficult to speculate in which direction or the magnitude of impact undetected infections would have had on apparent VE.

We have added this in the discussion.

2. Furthermore, you report a much lower vaccine efficacy in this study compared to previous studies, and I think this could be expanded on a little in the discussion. Why do you believe that vaccine efficacy (no prior infection) was so much lower- was this because you looked at it in isolation? And is this different to how other studies have looked at vaccine efficacy? And what effect does the fact you've looked HCW (with repeated exposures) have (if any_? )

Authors` response: Our population is provided health care free of cost and hence we were able to obtain clear datasets of health care workers who were vaccinated vs unvaccinated and a documentation of every HCW infection in our system. Therefore, we were indeed able to tease out/isolate the effect that a previous infection could have on acquisition on subsequent infection in vaccinated vs unvaccinated individuals. 

In addition, we have looked at the protective efficacy of vaccination during the epidemic caused by the delta variant which could have contributed to lower efficacy as well. Lower vaccine efficacy in infections due to the delta variant has been shown in other studies as well. 

Minor points: 

3. I thought the methods were very well written and clear. I just found the description of the cohort study a little confusing at times understanding the start and end dates. From my understanding you had a cohort between 16th April to 30th/31st May (second wave) 2021 and included all HCW. You then looked at infection events during this wave. Might be useful to have a sentence summing this up at the end of your descriptions of the cohort. (Also, minor point you mention 30th May in abstract but 31st in methods). 

Authors` response: We thank you for pointing out the error. We have done the changes. We have attempted to clarify this in the manuscript. 

4. I think in Supplementary table 1 it should maybe say "prior infection" rather than infection, and it could be a little confusing. I also think the reference group (which I believe is the unvaccinated, no prior infection) should be clearly stated - perhaps below the table. 

Authors` response: We have done the changes.

5. I think there's a mistake in Supplementary table 1 - the 95% confidence intervals for "vaccinated infected" group are smaller than the hazard ratio? 

Authors` response: We thank you for pointing out the error. The hazard ratio is 0.09 (95 % CI 0.05-0.159). We have corrected it in the manuscript.

6. There are some details missing from the methods (which are mentioned in the discussion) but I think it would help to have them in the methods:

• Type of testing done, and a clearer explanation of who was eligible (symptomatic and if you have a positive household contact?

• Vaccine type - To be added in the methodology section

• Previous infection (what was classified as a previous infection) 

Authors` response: As suggested by the reviewer, we have added the above details in the methods section.

---

## [Decision Letter · Decision Letter 1]

24 Mar 2022

PONE-D-21-35055R1Protective effect conferred by prior infection and vaccination on COVID-19 in a Healthcare Worker Cohort in South IndiaPLOS ONE

Dear Dr. Rupali,

Thank you for submitting your manuscript to PLOS ONE. After careful consideration, we feel that it has merit but does not fully meet PLOS ONE’s publication criteria as it currently stands. Therefore, we invite you to submit a revised version of the manuscript that addresses the points raised during the review process.

Your revised manuscript was reviewed by the two original experts in field. Although one reviewer was satisfied with the suggested modifications, the other identified a few remaining very important problems that require your careful attention. Please review the attached comments and provide point-by-point responses.

We look forward to receiving your revised manuscript.

Kind regards,

Yury E Khudyakov, PhD

Academic Editor

PLOS ONE

Reviewers' comments:

Reviewer's Responses to Questions

**Comments to the Author**

1. If the authors have adequately addressed your comments raised in a previous round of review and you feel that this manuscript is now acceptable for publication, you may indicate that here to bypass the “Comments to the Author” section, enter your conflict of interest statement in the “Confidential to Editor” section, and submit your "Accept" recommendation.

Reviewer #1: (No Response)

Reviewer #2: All comments have been addressed

2. Is the manuscript technically sound, and do the data support the conclusions?

Reviewer #1: Partly

Reviewer #2: Yes

3. Has the statistical analysis been performed appropriately and rigorously? 

Reviewer #1: No

Reviewer #2: Yes

4. Have the authors made all data underlying the findings in their manuscript fully available?

Reviewer #1: No

Reviewer #2: No

5. Is the manuscript presented in an intelligible fashion and written in standard English?

Reviewer #1: Yes

Reviewer #2: Yes

6. Review Comments to the Author

Reviewer #1: Following the initial review the authors corrected a good deal of criticism. However several points remain which are, for me, crucial to be addressed:

- the data was not made available, NOT EVEN TO THE REVIEWERS because the link the authors provide is not working: when trying to access it I get "DOI Not Found" the error; let me be very clear about this: if the next time the link is still not working I will be rejecting the paper for good. Here is the link provided by authors : Dryad - https://doi.org/10.5061/dryad.n8pk0p2x8

- there are severe limitations of the study because of the age homogeneity (everything concerns people with similar ages); these limitations should be VERY CLEARLY STATED : a) in the abstract, b) in the discussion section, c) when discussing the Cox regression, d) when discussing any generalization of the findings to another population.

Reviewer #2: (No Response)

7. PLOS authors have the option to publish the peer review history of their article (what does this mean?). If published, this will include your full peer review and any attached files.

Reviewer #1: No

Reviewer #2: No

---

## [Author Response · Author response to Decision Letter 1]

30 Apr 2022

Response to the Reviewers` Comments

Manuscript ID: PONE-D-21-35055

Reviewer #1: 

1. The data was not made available, NOT EVEN TO THE REVIEWERS because the link the authors provide is not working: when trying to access it I get "DOI Not Found" the error; let me be very clear about this: if the next time the link is still not working I will be rejecting the paper for good. Here is the link provided by authors : Dryad - https://doi.org/10.5061/dryad.n8pk0p2x8

Response:

The authors sincerely apologize for the trouble in the link. We have shared the private review link in the cover letter dated 18.02.2022. (https://datadryad.org/stash/share/fNoszj0lxymYnzlkSzW004Gu16B8UYYReQn8yqM6GSo) This dataset has been submitted to Dryad on 18.02.2022 and it was under the curation process. The DOI mentioned would have been activated only after acceptance of article, curation of the dataset and payment to the data repository. We have requested for an expedited data curation and we are happy to share that the data is now published on 17.04.2022. The variables which are considered as direct and indirect identifiers were removed as mandated by the data repository requirements. We are happy to share the entire dataset (excel) to the reviewer by email, if needed.

Link to access the dataset: https://doi.org/10.5061/dryad.n8pk0p2x8

2. There are severe limitations of the study because of the age homogeneity (everything concerns people with similar ages); these limitations should be VERY CLEARLY STATED : a) in the abstract, b) in the discussion section, c) when discussing the Cox regression, d) when discussing any generalization of the findings to another population.

Response:

The age distribution in our cohort varies mainly from 18-85 years of age which we feel is a fairly good distribution for HCW [Ref supplementary figure 1 which was provided on suggestion from the reviewer] and therefore we do not consider it as a severe limitation. The retirement age is 60 in our hospital and few employees contribute voluntary work after the retirement age (even upto 85 years). The age range is in keeping with other HCW studies in our country as we do have a younger work force when compared to western population (Eg SIREN study [1] which report a median age 45·7 years [IQR 35·4–53·5]).We do agree that median of 33.9 years of age does indicate a younger population but this is in keeping with the trend observed with COVID incidence in the general population in India (please note below)and hence we cannot categorize this as a severe limitation. Similar trends have been shown in studies published from India. 

A study by Kushwaha et al., 2021 [2] showed that the mean age of all COVID-19 patients was 39.47 ± 17.59 years and the difference between the mean age of males (39.98 ± 17.19 years) and females (38.50 ± 18.29 years) in a cohort of 112,860 covid positive patients in India. Another study published by Chatterjee et al., 2020 [3] reported that the mean age of the cases was 34.73 yr [±standard deviation (SD): 9.64; median: 33.0; interquartile range (IQR): 27-40] in a health care worker cohort. Another study by Reddy et al., 2021 aimed at looking the age difference between the first and second wave of pandemic in India by analyzing a total of 2,19,832 and 2,34,815 samples respectively [4]. The mean age during the first and the second wave were 35.1 ± 15.9 years and 46.1 ± 16.8 years respectively.

As suggested by the reviewer, we have clearly mentioned this as a possible limitation for generalization to the community at large in the “Limitations” section (Ref: page no 11; line number 287-289) and compared it with other studies in the discussion section (Ref: page no 10; line number 261-264). We do not think it merits mention in the abstract. 

References:

1. Hall VJ, Foulkes S, Charlett A, Atti A, Monk EJM, Simmons R, et al. SARS-CoV-2 infection rates of antibody-positive compared with antibody-negative health-care workers in England: a large, multicentre, prospective cohort study (SIREN). The Lancet. 2021;397: 1459–1469. doi:10.1016/S0140-6736(21)00675-9

2. Kushwaha S, Khanna P, Rajagopal V, Kiran T. Biological attributes of age and gender variations in Indian COVID-19 cases: A retrospective data analysis. Clin Epidemiol Glob Health. 2021;11: 100788. doi:10.1016/j.cegh.2021.100788

3. Chatterjee P, Anand T, Singh KhJ, Rasaily R, Singh R, Das S, et al. Healthcare workers & SARS-CoV-2 infection in India: A case-control investigation in the time of COVID-19. Indian J Med Res. 2020;151: 459–467. doi:10.4103/ijmr.IJMR_2234_20

4. Reddy MM, Zaman K, Mishra SK, Yadav P, Kant R. Differences in age distribution in first and second waves of COVID-19 in eastern Uttar Pradesh, India. Diabetes Metab Syndr. 2021;15: 102327. doi:10.1016/j.dsx.2021.102327

---

## [Editor Report · Decision Letter 2]

9 May 2022

Protective effect conferred by prior infection and vaccination on COVID-19 in a Healthcare Worker Cohort in South India

PONE-D-21-35055R2

Dear Dr. Rupali,

We’re pleased to inform you that your manuscript has been judged scientifically suitable for publication and will be formally accepted for publication once it meets all outstanding technical requirements.

Kind regards,

Yury E Khudyakov, PhD

Academic Editor

PLOS ONE
---

## [Editor Report · Acceptance letter]

12 May 2022

PONE-D-21-35055R2 

Protective effect conferred by prior infection and vaccination on COVID-19 in a healthcare worker cohort in South India 

Dear Dr. Rupali:

I'm pleased to inform you that your manuscript has been deemed suitable for publication in PLOS ONE. Congratulations! Your manuscript is now with our production department. 

Kind regards, 

on behalf of

Dr. Yury E Khudyakov 

Academic Editor

PLOS ONE